# Algorithm for Determining the Optimal Weights for the Akushsky Core Function with an Approximate Rank

Egor Shiriaev [1],* , Nikolay Kucherov [1] , Mikhail Babenko [1,2] , Vladislav Lutsenko [3] and Safwat Al-Galda [4]

1. Faculty of Mathematics and Computer Sciences Named after Prof. Nikolay Chervyakov, North-Caucasus Federal University, 355017 Stavropol, Russia; nkucherov@ncfu.ru (N.K.); mgbabenko@ncfu.ru (M.B.)
2. Control/Management and Applied Mathematics, Ivannikov Institute for System Programming, 109004 Moscow, Russia
3. North Caucasus Center for Mathematical Research, North-Caucasus Federal University, 355017 Stavropol, Russia; vvlutcenko@ncfu.ru
4. Mathematics Department, Faculty of Education, University of Misan, Amarah 62001, Maysan, Iraq; safwat.cj@gmail.com
* Correspondence: eshiriaev@ncfu.ru

**Abstract:** In this paper, a study is carried out related to improving the reliability and fault tolerance of Fog Computing systems. This work is a continuation of previous studies. In the past, we have developed a method of fast operation for determining the sign of a number in the Residue Number System based on the Akushsky Core Function. We managed to increase the efficiency of calculations by using the approximate rank of a number. However, this result is not final. In this paper, we consider in detail the methods and techniques of the Akushsky Core Function. During research, it was found that the so-called weights can be equal to random variables. Based on the data obtained, we have developed a method for determining the optimal weights for the Akushsky Core Function. The result obtained allows you to obtain a performance advantage due to the preliminary identification of optimal weights for each set of moduli.

**Keywords:** Residue Number System; Akushsky Core Function; Monte Carlo method; Fog Computing; Chinese Remainder Theorem

## 1. Introduction

In this paper, we continue the research described in the work [1]. In the presented work, a fast method for determining the sign of a number was developed based on the Akushsky Core Function (ACF). The developed method was based on the calculation of the approximate rank of a number. However, after exploring the basics of ACF, it was found that additional techniques can be applied.

The calculation of the approximate rank really makes it possible to increase the efficiency of calculating operations in the Residue Number System (RNS). This is confirmed by several scientific papers that have explored the different applicabilities of the rank of the RNS number. For example, in the article [2], RNS properties are used when developing a sign detection function for homomorphic encryption. RNS is used here to speed up the arithmetic of homomorphic encryption, which also allows the use of RNS techniques such as the rank of a number and its positional characteristic. In another article [3], the polynomial form of RNS (PRNS) is used to improve the reliability of cloud storage. Using PRNS as well as the entropy paradigm, the authors present a method to increase reliability by increasing the fault tolerance of the system due to the self-correcting properties of RNS. The result, according to the authors, allows you to correct errors as well as hardware and software failures. In addition, the presented method allows you to deal with integrity violations and the consequences of attacks and intrusions into the system. In the article [4], research is being undertaken related to the rank of the RNS number. The work presents

proof that a more efficient calculation of the rank is possible based on approximate methods. In addition, the authors demonstrate that the rank value can be obtained with the necessary accuracy.

Thus, this study is focused on a more detailed consideration of ACF within RNS, as well as its mathematical features and the possibility of applying them in practice.

We are continuing research aimed at improving the reliability of Fog Computing (FC) [5], which will allow more widespread use of FC technology. Fog computing (FC) is a way of organizing distributed computing based on interconnected low-power (edge) devices. Typical examples of FC are Smart City (SM) [6] and Internet of Things (IoT) [7] networks, where several low-power devices consist of various sensors (for example, humidity, temperature, light level, etc.), as well as traffic lights or, for example, mobile devices and on-board computers of cars. Thus, these low-power devices in large numbers create a computing network capable of processing a certain data stream, for example, creating the computing power necessary for the self-service needs of the IoT network SM.

Given the limited computing power of a single device, certain requirements are imposed on the network architecture.

Various methods of orchestration and load balancing allowed for increased efficiency in the architecture of FC networks. For example, in [8–10], the authors study orchestration methods for FC. There are also studies on load balancing in FC; for this, you can read the following works [11–15]. This made it possible to increase the efficiency of deploying the data transmission infrastructure. Reliability is affected by optimal load distribution between nodes and data transmission channels. However, in this case, the reliability is increased by manipulating the 2–7 levels of the OSI model. However, the first level, the physical level, is not considered by such methods.

At the physical level, reliability is determined by the fault tolerance of both the equipment and the information presentation system. RNS here demonstrates its effectiveness due to the properties of self-correction, which allow neglecting errors and anomalies of the physical layer up to a certain threshold. In addition, the case of FC-RNS demonstrates efficiency due to its natural parallelism.

However, RNS has a certain number of disadvantages associated with the computational complexity of non-modular operations, which is quite critical in the case of FC. To overcome this barrier, it is necessary to develop efficient and fast methods for computing non-modular operations in RNS. In the previous work, we concentrated on one of the most important operations of any number system, namely, determining the sign of a number. The developed method has shown its effectiveness. In this work, we set ourselves the goal of improving the characteristics obtained for this method and for ACF.

To do this, we conduct a detailed study of the main properties of ACF, namely, the so-called weights of the function. The study consists of finding the optimal weights that will reduce the computational complexity of operations with the required level of accuracy. To find the optimal weights, we use probabilistic methods, namely the Monte Carlo method, after which, using the developed algorithm, the optimal set is selected according to the specified parameters. These parameters are the number of RNS moduli in the set as well as their dimensions in bits.

The results obtained during the work make it possible to reduce computational costs when implementing methods and algorithms using RNS. We also show the positive impact of stochastic optimization methods on ACF; they allow you to effectively discover optimal weights among random values for ACF. In addition, we propose a method for obtaining optimal weights, which is implemented in the form of four moduli and presented in the work in the form of algorithms.

Thus, this work consists of the following sections: Section 2 discusses the basic concepts and characteristics of RNS and ACF; Section 3 describes the prerequisites, justification, and study itself, as well as the resulting algorithm; Section 4 presents the research results and their discussion; and Section 5 presents the results of the research work carried out as well as plans for further research.

## 2. About the Residue Number System and Akushsky Core Function

The Residue Number System (RNS) is a non-positional number system used to represent and process integers [16]. The main properties of RNS were presented in the Chinese Remainder Theorem as a solution to the system of linear comparisons modulo [17]. RNS received its modern representation in the works of Akushsky [18] and Garner [16]. RNS can also be thought of as performing modular arithmetic on a residue ring. To represent the number $X$ in RNS, it is necessary to obtain a set of relatively prime numbers $\{p_1, p_2, \ldots, p_n\}$, which are called RNS moduli. Then, you need to obtain the remainder by dividing $X$ by each modulo from the set, resulting in a set of remainders $\{x_1, x_2, \ldots, x_n\}$. Thus, RNS is a system of comparisons.

$$x_1 \equiv X(mod\ p_1),$$
$$x_2 \equiv X(mod\ p_2),$$
$$x_n \equiv X(mod\ p_n),$$

whose solution is the original number.

This representation of the number has certain advantages. Since a sufficient condition for a correct mapping of a number in RNS is that $X \in [0, P)$, where $P = \prod_{i=1}^{n} p_i$—is the RNS basis. To return a number to the positional system, you must use the following formula:

$$X = \left| \sum_{i=1}^{n} x_i \cdot B_i \right|_P \tag{1}$$

where $B_i$—orthogonal basis and $B_i = \left| P_i^{-1} \right|_{p_i} \cdot P_i$ where $P_i = \frac{P}{p_i}$, $\left| P_i^{-1} \right|_{p_i}$—inverse multiplicative element of $P_i$ modulo $p_i$.

We can represent a number up to 128 bits in four 32 bits, or eight 16 bits, etc. This allows you to control the limit on the size of numbers on 32/64 bit systems. In addition, each pair of remainders $x_i$ and modulus $p_i$ is independent of the others. This allows information to be processed in parallel. In addition, it allows you to use the self-correcting properties of the system.

In the 1970s, after describing the basic principles of RNS, Akushsky proposed a special function for a given number system based on the positional characteristic of a number, calling it the core function, or the Akushsky Core Function (ACF) [19]. During their research, Akushsky and Yuditsky, in parallel with the study of the rank of a number, proposed another positional characteristic of a number, the so-called core of a number. In articles [19,20], the considered core was presented.

Investigations of the core function originate from the so-called Lagrange branch [21] of the low positional system [22], based on the following comparison:

$$\lambda_1 x_1 + \lambda_2 x_2 + \cdots \lambda_n x_n \equiv W(mod\ D) = rD + W,$$

where $D$—low position system modulo, $r$—number rank, $\lambda$—system coefficients, and $W$ or $w_i$—weights.

Studies have shown that these weights can take on various values, both negative and greater than the modulus $D$ itself—which, by analogy with $P$, is a constraint. Further research by Akushsky showed that certain restrictions can be set for the values of $W$. The main characteristic of this function is the so-called range core $C(P)$. This value is dynamic and, like the number $X$, must be in the range $X \in \overline{1, P}$. In [20], Akushsky and Yuditsky established that, to simplify the practical implementation of the core function, $C(P)$ is equal to the largest modulo from the set, or their product.

Then, if $0 < w_i < C(P)$, then the core function is monotonically increasing, which expands the range of applicability of the core function. To determine the positional char-

acteristic of a number represented in RNS using ACF, it is necessary to introduce such a concept as the core of an orthogonal basis—$C(B_i)$. Which is located as follows:

$$C(B_i) = B_i \cdot \frac{C(P)}{P} - \frac{w_i}{p_i},$$ (2)

Then you can obtain the positional characteristic using the following formula:

$$C(X) = \left| \sum_{i=1}^{n} x_i \cdot C(B_i) \right|_{C(P)}.$$ (3)

Formula (3) has similarities with Formula (1). However, due to modulo division by the number $C(P)$, this method of calculating the core function of a number is not effective. In the same work, Akushsky proposed a different calculation option:

$$C(X) = \left( \sum_{i=1}^{n} x_i \cdot C(B_i) \right) - r_x C(P)$$ (4)

In this case, we replace the rather computationally complex operation of modulo division with multiplication with subtraction. If we consider Formula (4) critically, then we can note that determining the rank of a number is also a computationally complex and non-modular operation. However, in [1], our research group obtained a result from the calculation of the approximate rank of the required accuracy. The ACF itself is remarkable in that it allows you to determine the positional characteristics of the number represented in the RNS. This positional characteristic shows where on the number line the number being studied is located. If with a positional system we can explicitly evaluate a number, for example, is it greater than 0 or less, then in the case of RNS, this can be undertaken by returning the number to the positional system. This is where ACF comes to the rescue. Thus, ACF allows you to reduce the computational complexity of non-modular operations (for which you need to know where a number is on a line).

Additionally, it is worth noting that, in this case, we get only the positional characteristic of the number and not its real value. This allows you to determine the sign of a number, compare numbers, and perform various other operations.

ACF is considered an inefficient algorithm since the calculations still involve $P$, which makes it less efficient than, for example, the approximate method. However, this method has several positive properties that can be exploited, which will be presented in the next section.

## 3. A Probabilistic Approach to Determining the Optimal Weight of the Akushsky Core Function

Akushsky, in his writings on the weak positional system, RNS, and ACF, pointed out several different properties of RNS. Based on the fact that any number can be represented as a polynomial [23], Akushsky conducted a study of a weakly positional system based on the Lagrange formula, obtaining an interpolation polynomial, the components of which, expanded in a Taylor series, can perform their arithmetic. Adding to this the provisions of CRT, Akushsky obtained ACF, in fact, by discretizing the function of the so-called Lagrangian branch of the weakly positional system, that is, the interpolation polynomial.

Moreover, Akushsky pointed out [19] that the weights $w_i$ are specially chosen integers that can be obtained as follows:

$$w_i = \left| \left| P_i^{-1} \right|_{p_i} \cdot C(P) \right|_{p_i}$$ (5)

moreover, without noting the obligation of this calculation, only the limit of permissible values indicated above can be used.

Thus, given the fact that ACF is essentially a discretized function of the interpolation polynomial, $w_i$ can be given randomly.

Let us look at this with an example:

Take RNS with the following parameters: $x = (0.1, 6, 2)$, $p = (3, 5, 7, 11)$.

Let us calculate its parameters:

$P = 1155$, $P_1 = 385$, $P_2 = 231$, $P_4 = 105$, $\left|P_1^{-1}\right|_{p_1} = 1$, $\left|P_2^{-1}\right|_{p_2} = 1$, $\left|P_3^{-1}\right|_{p_3} = 2$, $\left|P_4^{-1}\right|_{p_4} = 2$, the weights obtained by Formula (5) have the form $w_1 = 2$, $w_2 = 1$, $w_3 = 1$, $w_4 = 0$.

We also take randomly generated weights $w_1' = 3$, $w_2' = 0$, $w_3' = 6$, $w_4' = 2$.

Calculate $C(B_i)$ and $C'(B_i)$ by Formula (2) and obtain $C(B_1) = 3$, $C(B_2) = 2$, $C(B_3) = 3$, $C(B_4) = 2$. $C'(B_1) = 2.6$, $C'(B_2) = 2.2$, $C'(B_3) = 2.28$, $C'(B_4) = 1.8$.

Then, calculating $X$ using the formula [24]:

$$X = \left| \frac{P}{C(P)} \cdot \left( \left| \sum_{i=1}^{n} x_i \cdot C(B_i) \right|_{C(P)} + \sum_{i=1}^{n} \frac{w_i}{p_i} \cdot x_i \right) \right|_P,$$

We obtain values $X = 321$ and $X' = 321 + 0.001 \cdot 10^{-2^{16}}$.

It is worth noting that the method of returning a number to a positional system based on ACF is extremely inefficient. In our study, we use it exclusively to identify errors.

The main disadvantage of most RNS algorithms is their high computational complexity. In a previous study, we were able to reduce the computational complexity of calculating the rank of a number by obtaining an approximation. In this case, we can consider the core function as an approximate positional characteristic of a number. Because the weights can be given randomly, we can find the optimal value of the weights $w_i$ at which the computational complexity of the operations will be reduced.

The weights $w_i$ participate in Formula (3) calculations. Then, we can reduce the computational complexity of the operations by finding the minimum value of the largest $C(B_i)$ or $minmax(C(B_1), C(B_2), \ldots, C(B_n))$.

To solve the problem, we can use the Monte Carlo methods [25]. Monte Carlo methods are a group of numerical methods aimed at studying random processes. This method was originally developed to solve physics problems related to neutrons in the 1940s. Now, these methods have gained popularity in various fields, including economics and mathematics. Our choice fell on this method due to the fact that it has a simple implementation and has proven itself to be a reliable and effective stochastic optimization method in the case of one random variable. In the future, we plan to conduct separate studies of the application of optimization methods for ACF, both stochastic and others.

We are interested in this method from the point of view of processing random weights. Having received a large sample of random weights, we can obtain data on the dependence of $C(B_i)$ from random ratios of weights, then process the obtained data and choose the optimal ratio.

We also chose the Monte Carlo method in terms of the positive qualities of ACF. A random spread of weights in the range $0 < w_i < C(P)$ imposes a small error on the result. As a method for determining the accuracy, we used the obtained values of the weights for the translation $(x_1, x_2, \ldots, x_n)$ into the positional system and calculated $|X - X'|$. Our studies have shown that the maximum error was $4 \cdot 10^{-2^{16}}$, which is a fairly accurate result.

The study proceeded as follows. The number of moduli and their size in bits were chosen. After that, the set $(p_1, p_2, \ldots, p_n)$ was generated as the first relatively prime numbers of a given length.

Then random weights were generated along the given boundary. Calculation of the set $(C(B_1), C(B_2), \ldots, C(B_n))$ and determination of the largest value from it. Returning a number to the positional system and calculating the absolute error. For each set of moduli, 10,000 sets of $w_i$ weights were generated. Further, for the convenience of processing, sorting in ascending order was carried out. From the bare minimum

$\max(C(B_1), C(B_2), \ldots, C(B_n))$, meanwhile, the values $(w_1, w_2, \ldots, w_n)$ and $|X - X'|$ sorted according to $\max(C(B_1), C(B_2), \ldots, C(B_n))$.

The process of determining the optimal set of weights consisted of two conditions: the value $\max(C(B_1), C(B_2), \ldots, C(B_n))$ should be minimal with the smallest error $|X - X'|$. Thus, this approach can be applied based on the requirements for the accuracy of the obtained values. For our study, we determined the maximum value of the error $0.1 \cdot 10^{-2^{16}}$. At the output, we obtained a set of weights that is optimal for a given set of moduli.

Based on the study, an algorithm for finding the optimal ACF weights was developed.

## 4. Method for Determining Optimal Weights

Based on the obtained data, a method was developed for selecting the optimal weights for ACF. The application of the method is effective in terms of speed. Since the RNS moduli are constant, it is possible to obtain a weight table at the stage of precomputation for a set of sets. The disadvantage is memory consumption, as the resulting weights must be stored.

Consider the algorithm of the obtained method. The method can be divided into several moduli:

1. The main modulo includes connections to all moduli. RNS initialization (Algorithm 1);
2. Constant calculation modulo calculates the constants of the selected RNS. For example, the basis of a set of moduli, multiplicative inversions, etc. (Algorithm 2);
3. Core function processing modulo calculates core function variables such as basis cores, orthogonal basis cores, etc. (Algorithm 3);
4. Monte Carlo modulo—searches for the optimal weight (Algorithm 4).

Consider modulo two.

---

**Algorithm 1**

---

**Input:** $p = \{p_1, p_2, \ldots, p_n\}$, X

---

**Output:** $w$

---

*Main*
1. **Calculation of** $P, \{P_1, P_2, \ldots, P_n\}, \left\{ \left| P_1^{-1} \right|_{p_1}, \left| P_2^{-1} \right|_{p_2}, \ldots, \left| P_n^{-1} \right|_{p_n} \right\}, \{B_1, B_2, \ldots, B_n\}$ **from** *Constant_function*
2. **Calculation of** *Error*, $w$, minmax $C(B)$ **from** *Monte_Carlo*
3. **print** $w$
**end**

---

**Algorithm 2**

---

**Input:** $p = \{p_1, p_2, \ldots, p_n\}$,

---

**Output:** $P, \{P_1, P_2, \ldots, P_n\}, \left\{ \left| P_1^{-1} \right|_{p_1}, \left| P_2^{-1} \right|_{p_2}, \ldots, \left| P_n^{-1} \right|_{p_n} \right\}, \{B_1, B_2, \ldots, B_n\}$,

---

*Constant_function*
1. **for** $i$ **in** $n$ **do**:
1.1. $P = P \cdot p_i$
2. **for** $i$ **in** $n$ **do**:
2.1. $P_i = \frac{P}{p_i}$
2.2. $\left| P_i^{-1} \right|_{p_i} = \text{mult\_inver}(P_i, p_i)$
2.3. $B_i = \left| P_i^{-1} \right|_{p_i} \cdot P_i$
**end**

---

$\text{mult\_inver}(P_i, p_i)$—performed according to the extended Euclid algorithm

---

Thus, the constants necessary for the operation of moduli three and four are calculated. Let us look at modulo three.

---

**Algorithm 3**

---

**Input:** $p = \{p_1, p_2, \ldots, p_n\}$, $x = \{x_1, x_2, \ldots, x_n\}$, $Constant\_function$, $w = \{w_1, w_2, \ldots, w_n\}$

---

**Output:** $C(B) = \{C(B_1), C(B_2), \ldots, C(B_n)\}$, $C(P)$, $X'$

---

*Core_func*
1. $C(P) = p_n$
2. **for** $i$ **in** $n$ **do**:
2.1. $C(B_i) = B_i \cdot \frac{C(P)}{P} - \frac{w_i}{p_i}$
3. **for** $i$ **in** $n$ **do**:
3.1 $C(X) + = x_i \cdot C(B_i)$
4. $C(X) = C(X) mod\, C(P)$
5. **for** $i$ **in** $n$ **do**:
5.1 $X' + = \frac{w_i}{p_i} \cdot x_i$
6. $X' = \left| \frac{P}{C(P)} \cdot (C(X) + X') \right|_P$
**end**

---

Now, after obtaining the necessary constants as well as a description of the calculation of the necessary core as well as the core function itself, we can describe modulo 4.

---

**Algorithm 4**

---

**Input:** $p = \{p_1, p_2, \ldots, p_n\}$, $x = \{x_1, x_2, \ldots, x_n\}$, $Constant\_function$, $Core\_func$, $X$, $Border$

---

**Output:** $Error$, $w$, minmax $C(B)$

---

*Monte_Carlo*
1. **for** $j$ **in** $Border$ **do**:
1.1. **for** $i$ **in** $n$ **do**:
1.1.1. $w_i^j = random(0, p_i)$
1.2. **Calculation of** $C(B_i)$ **from** $Core\_func$
1.3. $\max_j C(B) = C(B_1)$
1.3. **for** $i$ **in** $n$ **do**:
1.3.1 **if** $\max_j C(B) < C(B_i)$
1.3.1.1 $\max_j C(B) = C(B_i)$
1.4. **Calculation of** $X_j$ **from** $Core\_func$
2. **Sorting** * max $C(B)$ **with** $w, X$
3. **Cleaning** * max $C(B)$ **with** $w, X$
4. $Border = length\, of\, \max C(B)$
5. **for** $j$ **in** $Border$ **do**:
5.1. $Error = X - X'$
5.1 *if* $Error < 0.1 \cdot 10^{-2^{16}}$ **them**:
5.1.1 minmax $C(B) = \max_j C(B)$
5.1.2 $w = w^j$
5.1.3 **break**
**end**

---

sorting—ascending sort process max$C(B)$ with $w, X$
cleaning—destruction of duplicate values max$C(B)$ with $w, X$.

---

After describing additional moduli, we can move on to describing the main modulo.

Thus, these algorithms can be implemented as one program or used separately for other tasks not related to the topic of research.

Based on the described four moduli, a program was developed, based on which the results of Tables 1–3 were obtained.

**Table 1.** The results of the study on the size of the numbers.

| Length of Modulo | With Calculation Weight | Without Calculation Weight | With Optimal Weight |
|---|---|---|---|
| 8 | $3.03882 \cdot 10^{-5}$ | $2.95754 \cdot 10^{-5}$ | $2.94675 \cdot 10^{-5}$ |
| 16 | $4.54483 \cdot 10^{-5}$ | $4.51912 \cdot 10^{-5}$ | $4.45878 \cdot 10^{-5}$ |
| 32 | $7.62158 \cdot 10^{-5}$ | $7.56659 \cdot 10^{-5}$ | $7.49536 \cdot 10^{-5}$ |
| 64 | $1.45906 \cdot 10^{-4}$ | $1.45405 \cdot 10^{-4}$ | $1.44623 \cdot 10^{-4}$ |
| 128 | $3.0976 \cdot 10^{-4}$ | $3.08956 \cdot 10^{-4}$ | $3.08282 \cdot 10^{-4}$ |
| 256 | $7.5987 \cdot 10^{-4}$ | $7.58389 \cdot 10^{-4}$ | $7.57831 \cdot 10^{-4}$ |
| 512 | $2.140723 \cdot 10^{-3}$ | $2.136606 \cdot 10^{-3}$ | $2.135205 \cdot 10^{-3}$ |
| 1024 | $6.857154 \cdot 10^{-3}$ | $6.844715 \cdot 10^{-3}$ | $6.84191 \cdot 10^{-3}$ |

**Table 2.** Results of the study by the number of moduli in the set.

| Numbers in Set | With Calculation Weight | Without Calculation Weight | With Optimal Weight |
|---|---|---|---|
| 3 | $5.89298 \cdot 10^{-5}$ | $5.86213 \cdot 10^{-5}$ | $5.7904 \cdot 10^{-5}$ |
| 4 | $7.60817 \cdot 10^{-5}$ | $7.56052 \cdot 10^{-5}$ | $7.49824 \cdot 10^{-5}$ |
| 5 | $9.52133 \cdot 10^{-5}$ | $9.46818 \cdot 10^{-5}$ | $9.36631 \cdot 10^{-5}$ |
| 6 | $1.16094 \cdot 10^{-4}$ | $1.15005 \cdot 10^{-4}$ | $1.1437 \cdot 10^{-4}$ |
| 7 | $1.38983 \cdot 10^{-4}$ | $1.3774 \cdot 10^{-4}$ | $1.37104 \cdot 10^{-4}$ |
| 8 | $1.65344 \cdot 10^{-4}$ | $1.63639 \cdot 10^{-4}$ | $1.62737 \cdot 10^{-4}$ |
| 9 | $1.94102 \cdot 10^{-4}$ | $1.92349 \cdot 10^{-4}$ | $1.92099 \cdot 10^{-4}$ |
| 10 | $2.25351 \cdot 10^{-4}$ | $2.23501 \cdot 10^{-4}$ | $2.23485 \cdot 10^{-4}$ |
| 11 | $2.61388 \cdot 10^{-4}$ | $2.59014 \cdot 10^{-4}$ | $2.59359 \cdot 10^{-4}$ |
| 12 | $2.99725 \cdot 10^{-4}$ | $2.97459 \cdot 10^{-4}$ | $2.96376 \cdot 10^{-4}$ |
| 13 | $3.44273 \cdot 10^{-4}$ | $3.41763 \cdot 10^{-4}$ | $3.41997 \cdot 10^{-4}$ |
| 14 | $3.90702 \cdot 10^{-4}$ | $3.87631 \cdot 10^{-4}$ | $3.87106 \cdot 10^{-4}$ |
| 15 | $4.44745 \cdot 10^{-4}$ | $4.42496 \cdot 10^{-4}$ | $4.41398 \cdot 10^{-4}$ |
| 16 | $5.0257 \cdot 10^{-4}$ | $4.99252 \cdot 10^{-4}$ | $4.98557 \cdot 10^{-4}$ |
| 17 | $5.6871 \cdot 10^{-4}$ | $5.65295 \cdot 10^{-4}$ | $5.64622 \cdot 10^{-4}$ |
| 18 | $6.39716 \cdot 10^{-4}$ | $6.36057 \cdot 10^{-4}$ | $6.35331 \cdot 10^{-4}$ |
| 19 | $7.20255 \cdot 10^{-4}$ | $7.18 \cdot 10^{-4}$ | $7.16739 \cdot 10^{-4}$ |
| 20 | $8.08093 \cdot 10^{-4}$ | $8.01978 \cdot 10^{-4}$ | $8.01124 \cdot 10^{-4}$ |

**Table 3.** Research results.

| Number of Moduli | Modulo Size, Bit | Optimal Value $minmax(C(B_i))$ | Set of Optimal Weights | Absolute Error |
|---|---|---|---|---|
| 4 | 8 | 175.097276 | (257, 70, 171, 222) | $0.01 \cdot 10^{-2^{16}}$ |
| 4 | 16 | 45992.000091 | (52568, 29656, 26707, 65548) | $0.02 \cdot 10^{-2^{16}}$ |
| 4 | 32 | 3924937701.831088 | (2657113367, 4294721947, 1752687387, 729149879) | $0.01 \cdot 10^{-2^{16}}$ |
| 4 | 64 | 16095638220240922233.627427 | (18444693828608548770, 11556647366118817625, 12170730014670266392, 18387437696418261637) | $0.08 \cdot 10^{-2^{16}}$ |

**Table 3.** *Cont.*

| Number of Moduli | Modulo Size, Bit | Optimal Value $minmax(C(B_i))$ | Set of Optimal Weights | Absolute Error |
|---|---|---|---|---|
| 4 | 128 | 249716941177299120244684115304733596724.351813 | (340275175060926886165304188096319514004, 301020633665331798271476810118958979515, 336510905909786319187393834122600353777, 99612995849147588614905459610625716040) | $0.02 \cdot 10^{-2^{16}}$ |
| 4 | 256 | 84638047563640232122695814932807259328859751586507432993097330711003747665237.217685 | (115790836192975765166451773505058812555103554725275435478021644406706452852642, 113848405562115067747182346767019215315787742518548679960714055806362839975028, 104897699049763849793250581947950614534437641870821417673690586136930726447933, 98923094989414098966296758255814077133551266392072068398016455637357261600806) | $0.08 \cdot 10^{-2^{16}}$ |
| 4 | 512 | 11395397569604110152676168208487986321264231138783851769990239927836544809977810407709553641403814349725105916511666560766008823105913415076678588202679077.659230 | (1307379244356929620555957580182196503736664036227780907313385448381344149979018205855844166646873007363243247346391152622109823078619220482381715286142326, 5543164409286702716572601465622032585882845571826944323085098400619904321325605122924625864213659554356162432467580466006244982571591123246118185797630700, 1340760703016718623981837937193396281868600798045300349056754960848934881509072297516244238486375714475215395703895548338326362722714909449872398349270086553, 638325415687188403262431483587542085732706237832897015095684137130251257947650255142941524569524358882705633870825453557853910937588069767174410797637892) | $0.03 \cdot 10^{-2^{16}}$ |
| 4 | 1024 | 17390386487631557568877193356911449534797022649742650820875279939219721533684154987244752852440909216766456419992021748277878542083937871054015183914932458876690083274870869122592931606933916146527259690778373305228224144251618463272610700886448072724470899604614620546181257314179555871200198967252690396525188.328312 | (63671267529045939176782891807597164779330268199956543267364059858815426517192415527888751878361965987088021377634357435656840078061207715717709351178991647694721817028847289870083432365650336968494534407212371617155735472146154620756431111765526437519455533696878348787266959037481180284373151658835319232651, 4424725480713030677840622711187146930295559580169189326408788563584888382564703675326224130974169083513727872170871276596582715321580513170092484405352444947947823981492582592947297107287118473732837109986912937437269990827752644601820503739197964992645472095731613757167589033075081451503971427082800912256343, 17976612084570072772769235813661981360140887804560759723012786354441178092413793236878864506017795612020779621713172819656884659459971537103218335246165021703558837399335545006425174142144942717705542008919767347790879619910489640082914328259563779358224376662886767659247506099747141422184308331619563700807401, 142601023892529348692836223171335226264919772235731226363440967200738053328128603186683921762374795519794208008522994508516100471629894809907768831591019555964856820924572490471283932626475102107022108843426426778025198809114047631745504328467163232132797372534385677526604483471234849002200519138695152264587) | $0.1 \cdot 10^{-2^{16}}$ |
| 3 | 32 | 3568024123.214328 | (3837237931, 4294762609, 917896556) | $0.02 \cdot 10^{-2^{16}}$ |
| 4 | 32 | 3924937701.830790 | (2692113808, 4294901084, 2570217672, 1022682509) | $0.01 \cdot 10^{-2^{16}}$ |
| 5 | 32 | 3908695829.805370 | (4294390329, 3868346871, 2264663547, 928155071, 774470313) | $0.03 \cdot 10^{-2^{16}}$ |
| 6 | 32 | 3886001215.047813 | (3932482126, 4115409466, 1216882072, 1716628707, 4293640252, 2871721103) | $0.08 \cdot 10^{-2^{16}}$ |
| 7 | 32 | 4018445179.076804 | (4294885006, 2466465469, 391837946, 1166816806, 2142066795, 931058054, 3767377016) | $0.04 \cdot 10^{-2^{16}}$ |
| 8 | 32 | 3779516627.758768 | (3457587344, 1948271198, 4294064264, 3496188834, 736551483, 3109143057, 1682604, 2979033662) | $0.02 \cdot 10^{-2^{16}}$ |
| 9 | 32 | 3648257309.749618 | (3945387719, 3611636576, 3234580145, 3992778927, 1011101864, 1426654729, 4294456583, 696798307, 1248346682) | $0.03 \cdot 10^{-2^{16}}$ |

**Table 3.** *Cont.*

| Number of Moduli | Modulo Size, Bit | Optimal Value $minmax(C(B_i))$ | Set of Optimal Weights | Absolute Error |
|---|---|---|---|---|
| 10 | 32 | 3319015825.455541 | (1812568780, 1744305376, 2879177275, 2704532353, 740106217, 2133216454, 142907793, 1628758167, 4294755744, 3005613165) | $0.09 \cdot 10^{-2^{16}}$ |
| 11 | 32 | 3887252925.495630 | (4294843612, 4072774593, 3479907851, 298939644, 473835803, 2964747990, 567788892, 2441223305, 1589870977, 3410254302, 4125735668) | $0.007 \cdot 10^{-2^{16}}$ |
| 12 | 32 | 4116388674.763324 | (1505892188, 381403895, 2944743037, 1468923292, 872170926, 2727127099, 4294523003, 518813601, 2447761796, 2093868838, 642628657, 2306588867) | $0.03 \cdot 10^{-2^{16}}$ |
| 13 | 32 | 3839483959.395721 | (939171569, 3907368894, 64568480, 2218303052, 4293911622, 2393109237, 4186379519, 864348774, 2385043194, 2111471880, 1805406759, 1171095431, 1559353300) | $0.09 \cdot 10^{-2^{16}}$ |
| 14 | 32 | 3986159380.261221 | (616709959, 519448438, 477811205, 2399023990, 2083266316, 3919260116, 388017825, 2029968915, 3059987793, 4294573350, 2810153157, 3174088871, 2569069222, 285202218) | $0.04 \cdot 10^{-2^{16}}$ |
| 15 | 32 | 4033445816.904042 | (3720535967, 601463582, 555133089, 2930592128, 2232245455, 679510902, 108449824, 411127992, 3007487310, 1296564010, 612933647, 1775883604, 4294791549, 931800587, 3264057027) | $0.07 \cdot 10^{-2^{16}}$ |
| 16 | 32 | 3936456732.318885 | (187085904, 862570694, 3452682697, 768183615, 4209428887, 2440805784, 3193371456, 4294555286, 2837902995, 3705505493, 3771021744, 2723759171, 3695499094, 158931980, 2021526104, 348223685) | $0.03 \cdot 10^{-2^{16}}$ |
| 17 | 32 | 4252663218.916533 | (2119312200, 564715578, 18835399, 3613263029, 3832444712, 1661175387, 1545638111, 4294254100, 3133195295, 265247941, 3881549301, 1930557313, 1735739591, 1946257665, 103142023, 3119952405, 3245461581) | $0.08 \cdot 10^{-2^{16}}$ |
| 18 | 32 | 4277036539.883138 | (1721834367, 2312277614, 3273045534, 46432728, 1237032121, 1827774096, 1132197392, 2143658973, 3445768340, 1287239420, 2019238334, 1369430968, 1205953584, 1356879283, 1822651576, 3603025026, 4294935763, 2378100982) | $0.1 \cdot 10^{-2^{16}}$ |
| 19 | 32 | 4158164430.897782 | (1274653369, 2956041026, 1966664630, 3073660627, 409387091, 1296327460, 3559335394, 1490335973, 2302687431, 3834698696, 3180769032, 1210931162, 2414378075, 170084862, 4294240447, 3462515808, 3698892625, 1052828145, 3082019895) | $0.1 \cdot 10^{-2^{16}}$ |
| 20 | 32 | 4087064891.159399 | (2194625932, 3749491636, 738164928, 895890692, 1588777698, 3280055887, 1343047423, 3435587145, 1491029095, 2514950023, 2947973241, 4294680200, 3817426302, 659655741, 803902988, 4230645408, 638024259, 904912493, 1800410366, 1427397790) | $0.1 \cdot 10^{-2^{16}}$ |

The performance studies were carried out in a similar way to the previous study.

Research is conducted on the basis of programs written in Python, on equipment with the following characteristics:

- CPU: frequency: 2.90 GHz, cores—6, process technology: 14 nm;
- GPU: video memory 6144 MB, memory frequency 14000 MHz, GPU frequency 1680 MHz, TDP 500 W;
- RAM: 16 GB, frequency 3200 MHz;
- OS: Windows 11.
- The experiment is carried out in two stages:
- Stage A—performance study of 9 sets, 8 moduli, and dimensions from 8 to 1024 bits;
- Stage B—performance study of 20 sets, from 3 to 20 moduli, and a dimension of 32 bits.

When conducting a two-stage simulation, the time characteristics of each method were obtained. The results obtained are reflected in the figures (Figures 1 and 2), where is the absolute error equal to $0.1 \cdot 10^{-2^{16}}$, and the tables (Tables 1–3).

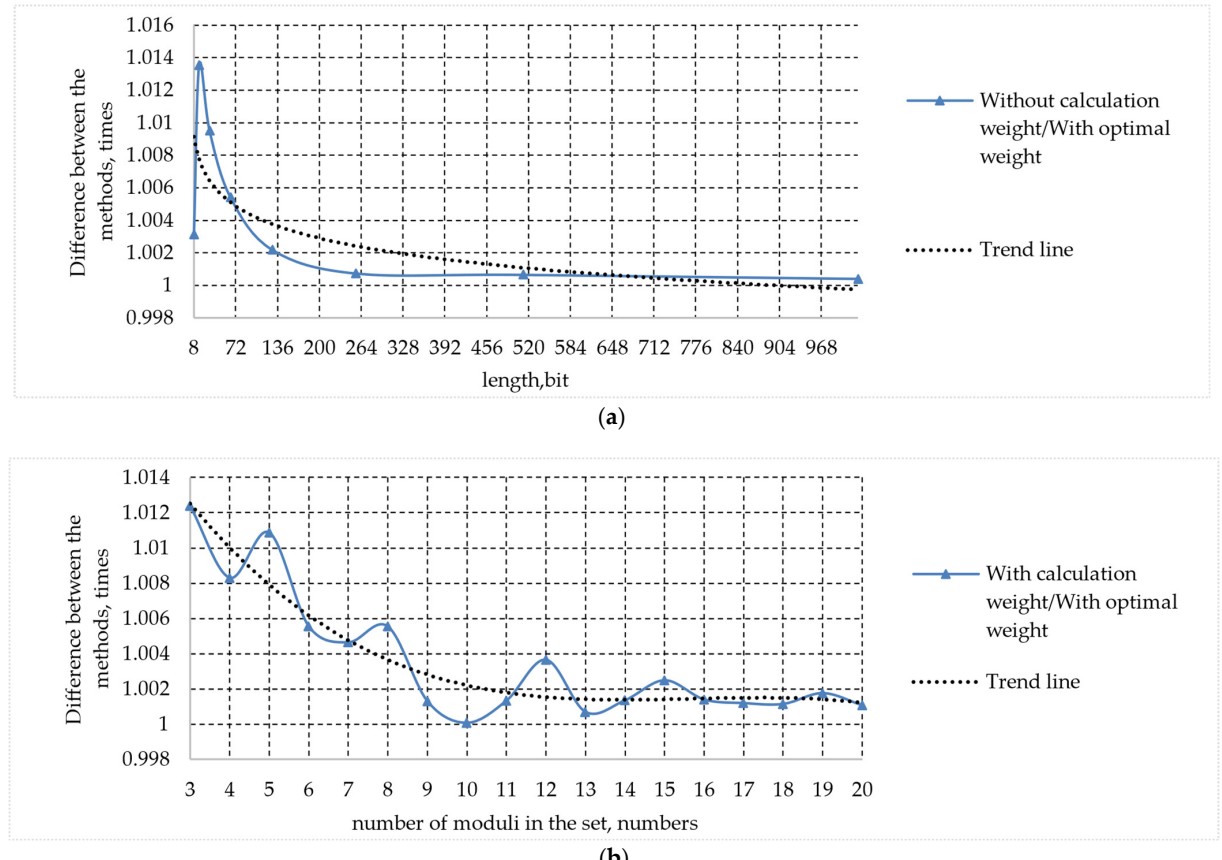

**Figure 1.** The results of the study: (**a**) stage A and (**b**) stage B.

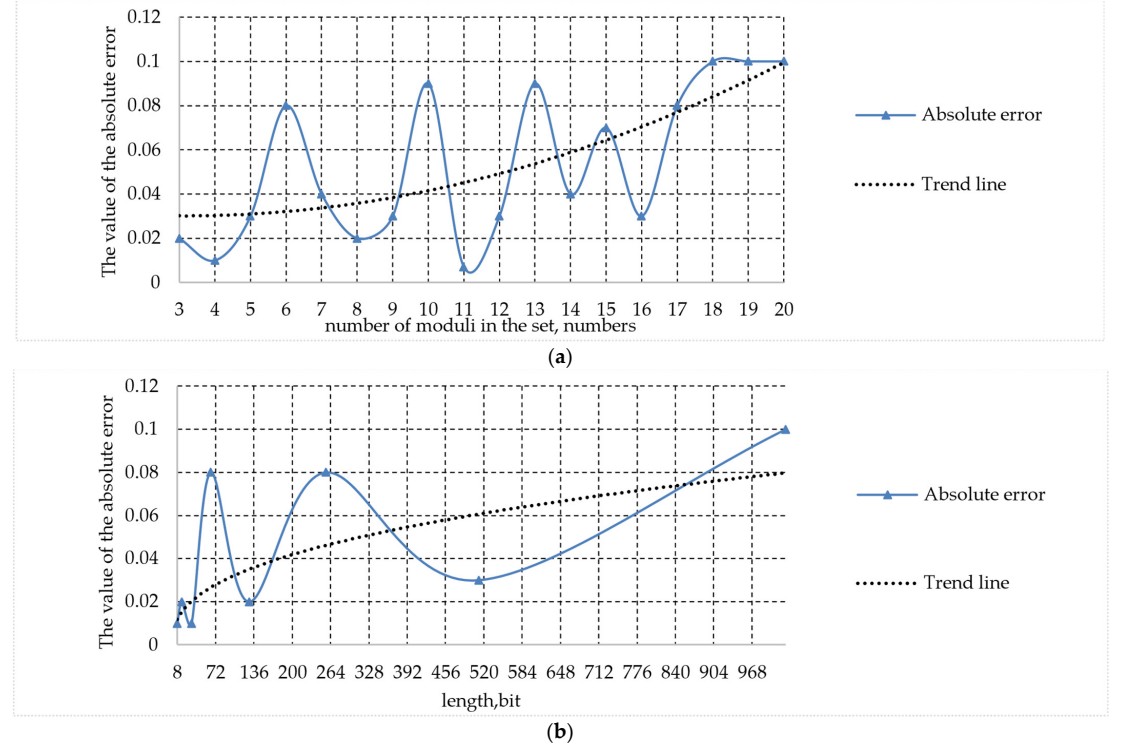

**Figure 2.** The results of the study: (**a**) stage A and (**b**) stage B.

Namely, sets of moduli and their numbers, as well as optimal weights, were taken. With these values, the positional characteristics of the number were calculated as $X$. The same was undertaken under the conditions of the weights obtained by Formula (5).

Based on which three-time characteristics were obtained:

1. Positional characteristics are obtained based on optimal weights;
2. Positional characteristics are obtained based on the calculated weights with the measurement of the time of their calculation;
3. Positional characteristics are obtained based on the calculated weights without measuring the time of their calculation.

Based on the experiment, the following tables were obtained.

Let us consider the results obtained. The results obtained are presented both as tables (Tables 1 and 2).

Based on the data obtained, we can say that the method we have developed for finding optimal weights is effective. Efficiency was confirmed in both cases. When weights are calculated and when weights are stored on disk. In both cases, the use of optimal weights makes it possible to reduce the computational complexity of determining the positional characteristics based on ACF. The result is explained by the reduction in the size of $C(B)$, which reduces the computational cost.

Analyzing the results of Table 2, we can say that the result obtained in Table 1 is adequate. With this study, we confirmed that the resulting sets are stable, both with an increase in the size of the moduli and with an increase in their number. Thus, it is possible to use both the method presented in the work and the obtained weight table in the operation of real systems. The benefit obtained in our study is within 1%. Despite the small productivity gains, this is an important result. When using RNS methods for the safety and reliability of systems, this increase is noticeable. For example, RNS is used in homomorphic encryption [26–28]. Homomorphic encryption arithmetic has high computational complexity. RNS is used to speed it up due to some of its properties. Thus, an increase of even 1% will allow the use of this type of encryption with greater efficiency.

We examine the data indicated in the table as follows. Since, based on the data, the performance advantage of our method is obvious, let us compare our method with the experiment closest in performance. Namely, with the classical calculation of the core without calculating the weights. Additionally, we obtain the following graphs (Figure 1). To do this, we divide the values without calculation weight by the optimal weight and obtain how many times our method is more effective than the classical one.

Here, we can observe the following picture. In general, throughout the study, the proposed method had an advantage. However, we can observe the following trend. Depending on the size of the moduli, the performance ratio of both methods tends to unity (Figure 1a) exponentially. This is explained by the fact that with an increase in the size of the moduli, the computational complexity of the calculations also increases. The essence of our method is to manage this complexity by introducing optimal weights. However, due to the fact that for huge values of moduli, the computational complexity is colossal, the influence of optimal weights on it is reduced. However, there is still relevance here. Special attention is paid to the performance value when the module size is 8 bits. This anomalous value (which is clearly less than expected) is explained by the low computational complexity of the ACF calculation operation. However, as we can observe, the performance is still higher since, in the presence of optimal weights, the value of $C(B_i)$ is less.

In the case of stage B, we can observe the following (Figure 1b). The performance ratio of the methods also tends to unity; however, we can observe that the graph line is broken. This can be explained by the fact that, although the number of moduli increases the computational complexity, it is not as fast as in the case of stage A. This is because in RNS, in most situations, only the number of addition operations in formulas depends on the number of moduli, which carries a lower computational load. In addition, as mentioned above, we use the Monte Carlo method based on random weights, which of course can also

have a certain effect on the result. However, as mentioned above, the proposed method had the best performance throughout the study.

As a result of the simulation, we obtained the following results, presented in Table 3.

Let us consider the received data in more detail. The first column stores the number of moduli in the set, and the second column stores the size of one modulo in the set. These columns are necessary for the convenience of finding the required set of weights. The third column, the value of $\max C(B)$, allows you to compare the basis on which the set of optimal weights was determined. The fourth column is a set of optimal weights. The fifth column displays the absolute error of converting a number from RNS to a positional system when using the resulting set of weights. Let us analyze the value of the absolute error. To do this, we constructed the graph in Figure 2.

In this case, we can observe the following. Despite some deviations, in both cases, the value of the error on average increases with the growth of the computational complexity of the experiment, which is a logical result. However, the value of the error does not go beyond the previously indicated boundary, which is equal to $0.1 \cdot 10^{-2^{16}}$, which is a good result.

## 5. Conclusions

This paper continues the research that was started in [1]. The work included a part of the study devoted to research related to the calculation of ACF. It has been found that the ACF weights can be generated randomly without seriously affecting the accuracy of the result.

Based on this fact, a method was developed to obtain the optimal set of weights for a given set of moduli. The method consists of applying the Monte Carlo method to iterate over random values of the weights to find the most appropriate result. As an optimum criterion, the following was chosen: the minimum–maximum value of the core of the basis $C(B)$, as well as the absolute calculation error equal to $0.1 \cdot 10^{-2^{16}}$.

Based on the method obtained, a performance study was conducted. The performance of the previous study was taken as a benchmark [1]. It also added the performance of calculating the positional characteristic without considering the calculation of weights, as well as the sets we offer. The results of the experiment allow us to talk about the effectiveness of the solution obtained based on the optimal ACF core.

In future studies, we plan to conduct even more experiments related to finding the optimal value of the ACF coefficients. As well as implementation and testing on real systems. In addition, we plan to conduct a more detailed study of optimization methods for ACF.

**Author Contributions:** Conceptualization, methodology, software, validation, research, manuscript—writing, E.S.; methodology, research, manuscript—writing, supervision, N.K.; conceptualization, methodology, research, manuscript—writing, supervision, M.B.; conceptualization, methodology, research, manuscript—writing, V.L.; methodology, validation, manuscript—writing, S.A.-G. All authors have read and agreed to the published version of the manuscript.

**Funding:** This work was supported by the Ministry of Education and Science of the Russian Federation (Project 075-15-2020-788).

**Institutional Review Board Statement:** Not applicable.

**Informed Consent Statement:** Not applicable.

**Data Availability Statement:** Not applicable.

**Conflicts of Interest:** The authors declare no conflict of interest.

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
