# Peer review of "Algorithm for Determining the Optimal Weights for the Akushsky Core Function with an Approximate Rank"

_applsci, doi:10.3390/app131810495_

Round 1

Reviewer 1 Report

The authors continued the work for Akushsky Core Function by developing a method based on Monte Carlo for determining the optimal weights. My concerns are listed in the following.

  1. Why do you choose Monte Carlo to optimize the weights? What is the difference for other optimization methods to optimize the weights of ACF?
  2. In Figure 1, the performance lines of the three variants are almost the same. I didn’t see a significant effect of different methods. The difference between different methods is also less than 1%. Please highlight the advantages of your algorithm.
  3. In table 1, why the performance of “without calculation weight” is better than “with calculation weight”?
  4. In Figure 2(a), why does the difference line go up first and then go down, aka, why there is a peak in the line?
  1. In all figures and tables, you wrongly spell “weight” by “weigth”.

Reviewer 2 Report

It is understood that this paper represents a step in an ongoing research, with its focus on improving efficiency of the necessry calculus load without degrading performance.

Given this scope, the paper is an acceptable and interesting presentation of results so far, with a useful and interesting insight in the methodoly used

However, the graphs in FIgure 1 and 2 are confusing:

- giving one graph for 3 options looks not credible

- indicating different colours and showing only one is not helpful

it is suggested to add tables showing the actual values; if the differences are snall enough then in a graph this may not be visible, but have only the graph representing 3 options leads to some suspicion

Reviewer 3 Report

The strength of this article is handling the base science field of computational complexity. It needs high-level abstract thinking and excellent knowledge of high mathematics.

1. The article is written with a high abstract. To give some numerical results in conclusion, what is the efficiency of using the Akushinsy Core Function?

2. Needs to give in the introduction the novelty of the article.

3. More explain the essence of terms in the article. Terms are used, but for other field readers, they are mostly unknown. To obtain more comprehensive target  readers need more explanations,

4. After the formula (10) is given, the same formula under the heading Lemma 1. Is Lemma 1 a specific term in this field or not? If we use Lemma 1, the readers are waiting for Lemma 2.

5. In row 225, have the statement about the maximum error value. From what suggestions is it stated?

6. In the 4-th chapter, given four algorithms. Not described what is algorithms' significance.

7. Figure 1 on page 8 needs to be clarified. We find three different calculation modes from the legend but only one in the figures.  Maybe change the scale of the figures or add additional explanations.

8. Using formulas needs to clearly show whether these are the property of the article authors or taken from references.
